# Dysbiosis of Gut Microbiota Contributes to Uremic Cardiomyopathy via Induction of IFNγ-Producing CD4+ T Cells Expansion

Bin Han,[a] Xiaoqian Zhang,[a] Ling Wang,[a] (ID) Weijie Yuan[a]

[a]Department of Nephrology, Shanghai General Hospital, Shanghai Jiaotong University School of Medicine, Shanghai, China

Bin Han and Xiaoqian Zhang contributed equally to this work. The order of authors was determined by their contributions.

**ABSTRACT** Uremic cardiomyopathy (UCM) correlates with chronic kidney disease (CKD)-induced morbidity and mortality. Gut microbiota has been involved in the pathogenesis of certain cardiovascular disease, but the role of gut microbiota in the pathogenesis of UCM remains unknown. Here, we performed a case-control study to compare the gut microbiota of patients with CKD and healthy controls by 16S rRNA (rRNA) gene sequencing. To test the causative relationship between gut microbiota and UCM, we performed fecal microbiota transplantation (FMT) in 5/6th nephrectomy model of CKD. We found that opportunistic pathogens, particularly *Klebsiella pneumoniae* (*K. pneumoniae*), are markedly enriched in patients with CKD. FMT from CKD patients aggravated diastolic dysfunction in the mouse model. The diastolic dysfunction was associated with microbiome-dependent increases in heart-infiltrating IFNγ+ CD4+ T cells. Monocolonization with *K. pneumoniae* increased cardiac IFNγ+ CD4+ T cells infiltration and promoted UCM development of the mouse model. A probiotic *Bifidobacterium animalis* decreased the relative abundance of *K. pneumoniae*, reduced levels of cardiac IFNγ+ CD4+ T cells and ameliorated the severity of diastolic dysfunction in the mice. Thus, the aberrant gut microbiota in CKD patients, especially *K. pneumoniae*, contributed to UCM pathogenesis through the induction of heart-infiltrating IFNγ+ CD4+ T cells expansion, proposing that a Gut Microbiota-Gut-Kidney-Heart axis could play a critical role in elucidating the etiology of UCM, and suggesting that modulation of the gut bacteria may serve as a promising target for the amelioration of UCM.

**IMPORTANCE** Uremic cardiomyopathy (UCM) correlates tightly with increased mortality in patients with chronic kidney disease (CKD), yet the pathogenesis of UCM remains incompletely understood, limiting therapeutic approaches. Our study proposed that a Gut Microbiota-Gut-Kidney-Heart axis could play a critical role in understanding etiology of UCM. There is a major need in future clinical trials of patients with CKD to explore if modulation of gut microbiota by fecal microbiota transplantation (FMT), probiotics or antibiotics can alleviate cardiac dysfunction, reduce mortality, and improve life quality.

**KEYWORDS** gut microbiota, IFNγ+ CD4+ T cell, uremic cardiomyopathy, chronic kidney disease, cardiomyopathy

Address correspondence to Weijie Yuan, weijie.yuan@shgh.cn.

The authors declare no conflict of interest.

Chronic kidney disease (CKD) is a leading public health issue that affects the global burden of mortality caused by cardiovascular disease (1–8). The major phenotype of cardiovascular comorbidities in individuals with CKD—uremic cardiomyopathy (UCM)—is best characterized as diastolic dysfunction seen in conjunction with left ventricular hypertrophy (9, 10) and predicts hospitalization and mortality in this patient

**TABLE 1** Characteristics of the participants included in this study[a]

| Characteristics | CKD patients (n = 122) | Healthy controls (n = 61) | P value |
|---|---|---|---|
| Demographics | | | |
| Age (yrs) | 57.9 ± 9.38 | 56.9 ± 3.65 | 0.443 |
| Female, n (%) | 49 (41.2) | 26 (42.6) | 0.750 |
| BMI (kg/m²) | 21.8 ± 5.36 | 22.3 ± 3.80 | 0.569 |
| Primary kidney disease | | | |
| Diabetes, n (%) | 36 (29.5) | NA | |
| Hypertension, n (%) | 33 (27.0) | NA | |
| Glomerulonephritis, n (%) | 29 (23.8) | NA | |
| Polycystic kidney, n (%) | 6 (4.9) | NA | |
| Other, n (%) | 18 (14.8) | NA | |
| CKD clinical stage | | | |
| Stage 3 | 3 (28.7) | NA | |
| Stage 4 | 41 (33.6) | NA | |
| Stage 5 | 46 (37.7) | NA | |

[a]Quantitative values are presented as mean ± SD, and qualitative values are presented as number of participants/percentage.

population (11–13). However, the underlying mechanisms of development of UCM are poorly defined and current treatments that are effective in other cardiomyopathy only modestly improve clinical outcomes in UCM (10).

T cells have been mechanistically linked not only to hypertensive and diabetic cardiomyopathy, but also to cardiac dysfunction and remodeling during chronic heart failure (14–18). In CKD, accumulation of $CD4^+$ T cells in the peripheral circulation has been strongly correlated with cardiovascular events in this patient population (19). A recent study by Pamela et al. identified increased $CD4^+$ T cells, especially $IFN\gamma^+$ $CD4^+$ T cells, in the heart tissue of CKD mice and demonstrated a causal role for the heart-infiltrating T cells in the pathogenesis of UCM (20).

Gut microbiota, which constitutes the largest microecosystem in the host body, plays a key role in modulation of host immunity and disease (21–23). Gut dysbiosis has been implicated in the pathogenesis of certain cardiovascular disease in a T cell-dependent manner (24) and fecal microbiota transplantation (FMT) or administration of certain gut commensals ameliorates cardiac dysfunction in mice with heart failure (25). Emerging studies have uncovered that patients with CKD had obvious gut dysbiosis characterized by enriched *Klebsiella pneumoniae* (*K. pneumoniae*) and other *Enterobacteriaceae* (e.g., *Escherichia*, *Shigella*, *Salmonella*) (26, 27). Members of *Klebsiella* are among the most common intestinal opportunistic pathogens and are believed to be strong inducers of $IFN\gamma^+$ $CD4^+$ T cells (28–30). Yet, the role of gut microbiome, particularly *K. pneumoniae*, in the pathogenesis of UCM remains to be elucidated. Considering the pathogenic role for cardiac $IFN\gamma^+$ $CD4^+$ T cells in UCM and the elevated abundance of *K. pneumoniae* in CKD, we hypothesized that the aberrant gut microbiota, *K. pneumoniae* in particular, would contribute to development of UCM via the induction of heart-infiltrating $IFN\gamma^+$ $CD4^+$ T cells expansion.

## RESULTS

**Characteristics of gut microbiota in the cohort and recipient mice.** A total of 183 participants were included in this study, including 122 individuals with stages 3 to 5 CKD and 61 healthy controls matched by gender, age, and body mass index (BMI) (Table 1). Among CKD patients, 36 (29.5%) had a current diabetes and their mean BMI was 22.0 ± 4.82 kg/m². By 16S rRNA gene sequencing, we found that fecal microbial composition and alpha-diversity of CKD patients differed from that of healthy controls (Fig. 1A and B). At the species or genus level, *K. pneumoniae* was markedly enriched in patients with CKD along with other commensals such as *Ruminococcus* spp., *Alistipes* spp., and

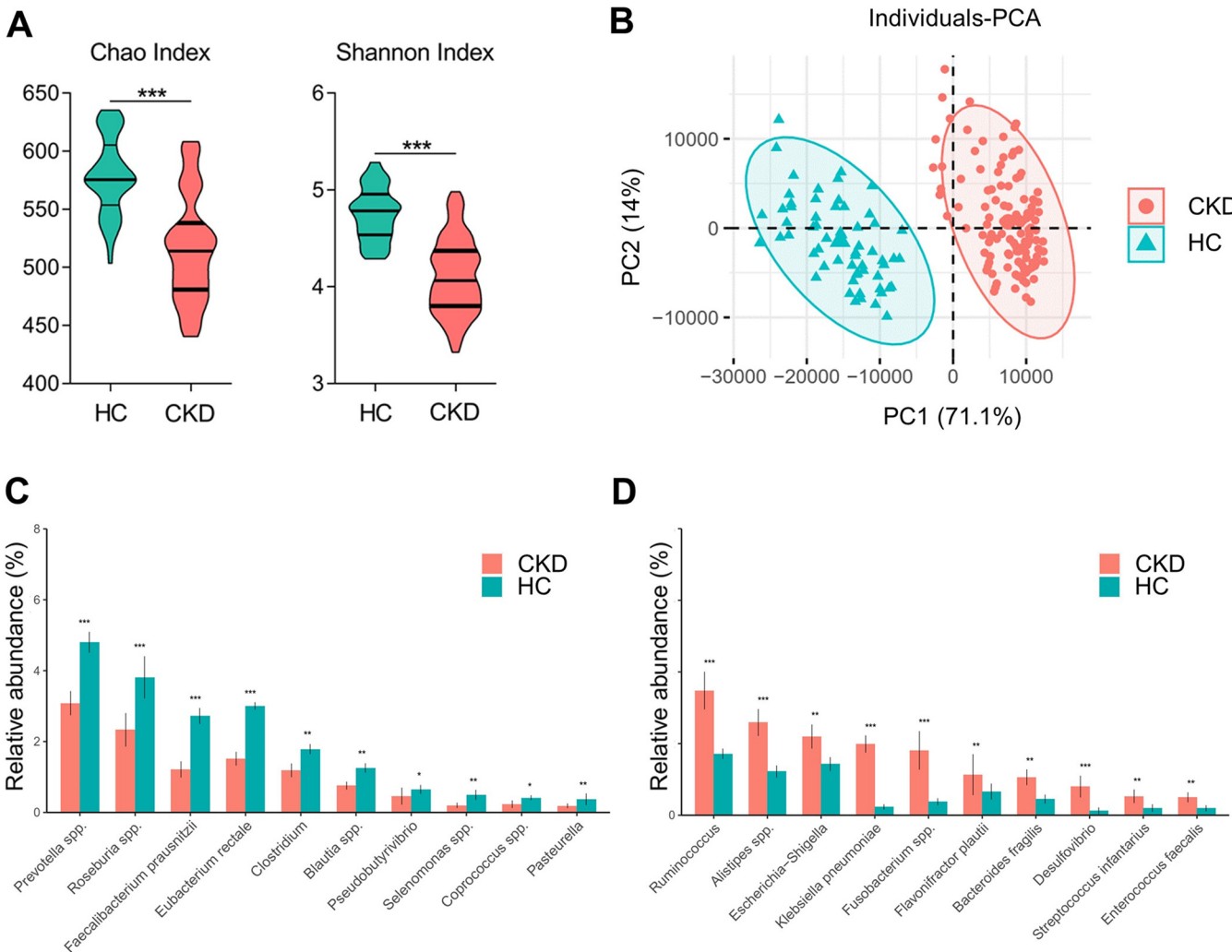

**FIG 1** Characteristics of gut microbiota in the cohort. (A) Comparison of alpha-diversity indices (Chao1 Index and Shannon Index) between CKD and HC groups (CKD: $n = 122$, HC: $n = 61$). (B) PCA based on OTUs distribution revealed that the gut microbial composition was significantly different between CKD and HC (CKD: $n = 122$, HC: $n = 61$). (C–D) Boxplot showed the prominent species that differ significantly in relative abundance between CKD and HC. Boxes denoted the interquartile range between the first and third quartiles. Whiskers represented the lowest and highest values within 1.5 times the range of the first and third quartiles (CKD: $n = 122$, HC: $n = 61$). *, $P < 0.05$; **, $P < 0.01$; ***, $P < 0.001$.

*Escherichia-Shigella*, with a relative underrepresentation of *Prevotella* spp., *Roseburia* spp., *Faecalibacterium prausnitzii*, *Eubacterium rectale*, and *Clostridium* (Fig. 1C and D). We transplanted the gut microbiota from either healthy controls or CKD patients into antibiotic treated microbe-depleted CKD mice (Fig. 2A). Oral administration of an antibiotic cocktail for 2 weeks after nephrectomy resulted in marked decrease of operational taxonomic units (OTUs). The donors were selected as having the highest levels of control-enriched and CKD-enriched species, respectively. After FMT, the abundance variations in *K. pneumoniae* showed the same trend between recipient mice and donor mice (Fig. 2B) and the recipient mice efficiently recaptured the gut microbial composition of the control or patient (Fig. 2C and D).

**Gut microbiota from CKD patients exacerbated diastolic dysfunction in the mice.** We found that CKD recipients developed more severe diastolic dysfunction compared to healthy controls (HC) recipients or control mice (Fig. 3A and B). There were no significant differences in terms of EF and diastolic left ventricle anterior wall thickness (LVAW;d) between the four groups studied (Fig. 3C and D). Notably, the relative abundance of *K. pneumoniae* was positively correlated with diastolic dysfunction based on Spearman correlation calculated for CKD recipients and HC recipients (Fig. 3E and H).

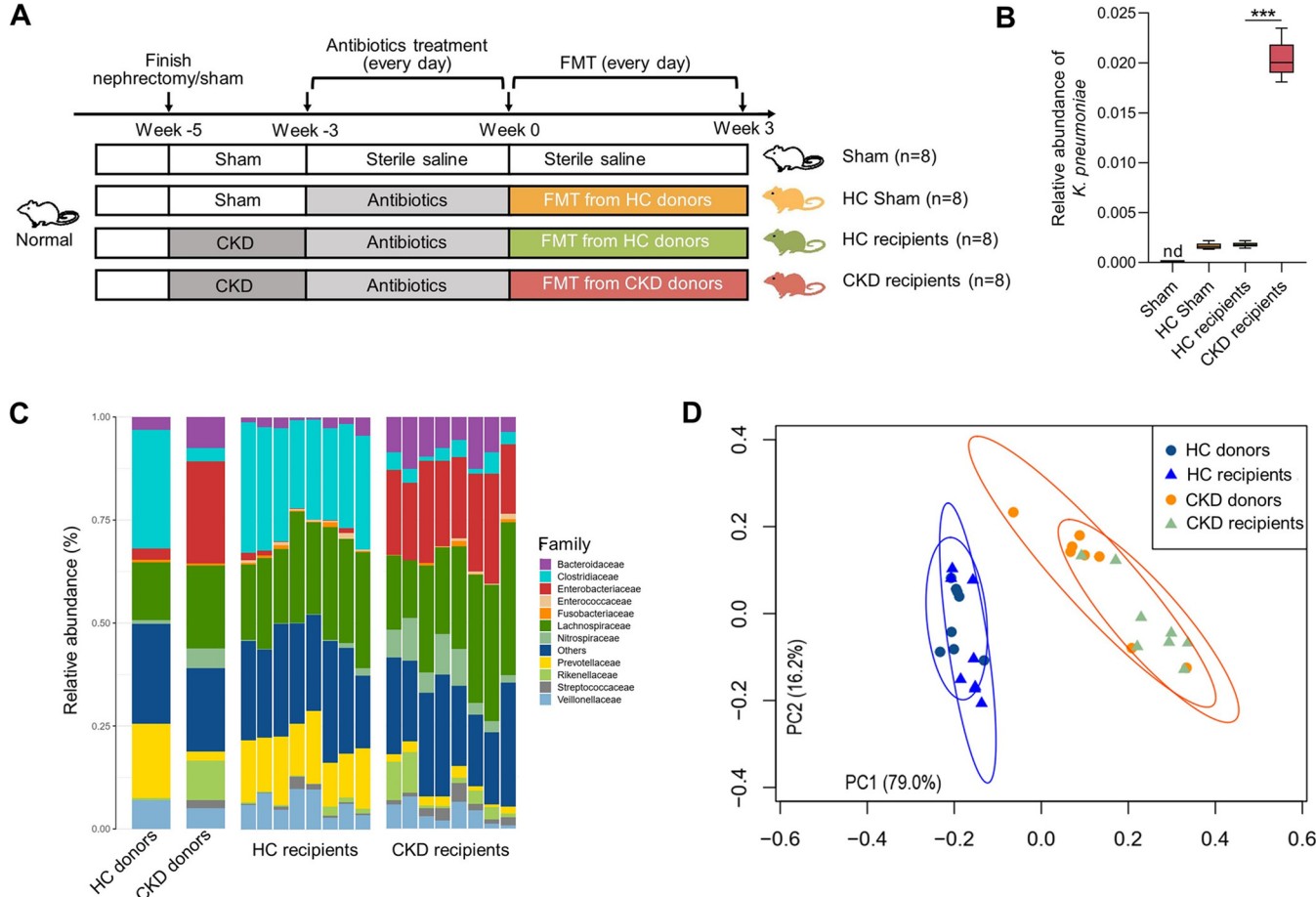

**FIG 2** Fecal microbiota transplantation experiment. (A) Schematic representation of FMT experiment (*n* = 8 per group). (B) As donors, the relative abundance of *K. pneumoniae* in fecal samples of CKD recipients was significantly higher than that of other recipients. Boxes and whiskers show mean ± interquartile range. Data were compared by Kruskal-Wallis test (*n* = 8 per group, *P* values were adjusted by the Bonferroni correction). Each bar denoted a group (X-axis) with mean and 95% CI (Y-axis). (C–D) Comparison of gut microbial taxa (C) and PCoA (D) showed no significant alternations of the gut bacterial communities of donors and recipients (*n* = 8 per group). *, *P* < 0.05; **, *P* < 0.01; ***, *P* < 0.001.

Additionally, no significant differences of systolic blood pressure (SBP) and kidney function were observed among CKD recipients and HC recipients (Fig. 3I and J). Collectively, these results indicate a causative contribution of the gut microbiota from patients with CKD to development of UCM in the mouse model.

**Gut microbiota from CKD patients induced expansion of heart-infiltrating IFNγ+ CD4+ T cells in the mice.** T cells, particularly IFNγ+ CD4+ T cells, play a causal role in diastolic dysfunction during UCM (20). Thus, we examined whether the aggravated diastolic dysfunction observed with FMT was correlated with changes in levels of IFNγ+ CD4+ T cells in the heart. To ascertain the presence of T cells in the heart tissue, we conducted immunofluorescence analysis. The infiltration of CD4+ T cells was clearly observed in the cardiac tissue of CKD recipients (Fig. 4A and B). Flow cytometric analysis of cardiac lymphocytes further revealed a significant increase in IFNγ+ CD4+ T cells infiltration in the heart tissue of CKD recipients (Fig. 4C). The activity of heart-infiltrating IFNγ+ CD4+ T cells (Fig. 4D), but not IL-17+ CD4+ T cells (Fig. 4E), depended on the gut microbial condition of the mice. Additionally, the relative abundance of *K. pneumoniae* was significantly correlated with levels of cardiac IFNγ+ CD4+ T cells based on Spearman correlation calculated for the recipients with CKD (Fig. 4F and G). No correlation was observed between abundance of other microbial taxa and levels of cardiac IFNγ+ CD4+ T cells. Thus, it is conceivable that the gut microbiota derived UCM, at least in part, via the induction of cardiac IFNγ+ CD4+ T cells expansion.

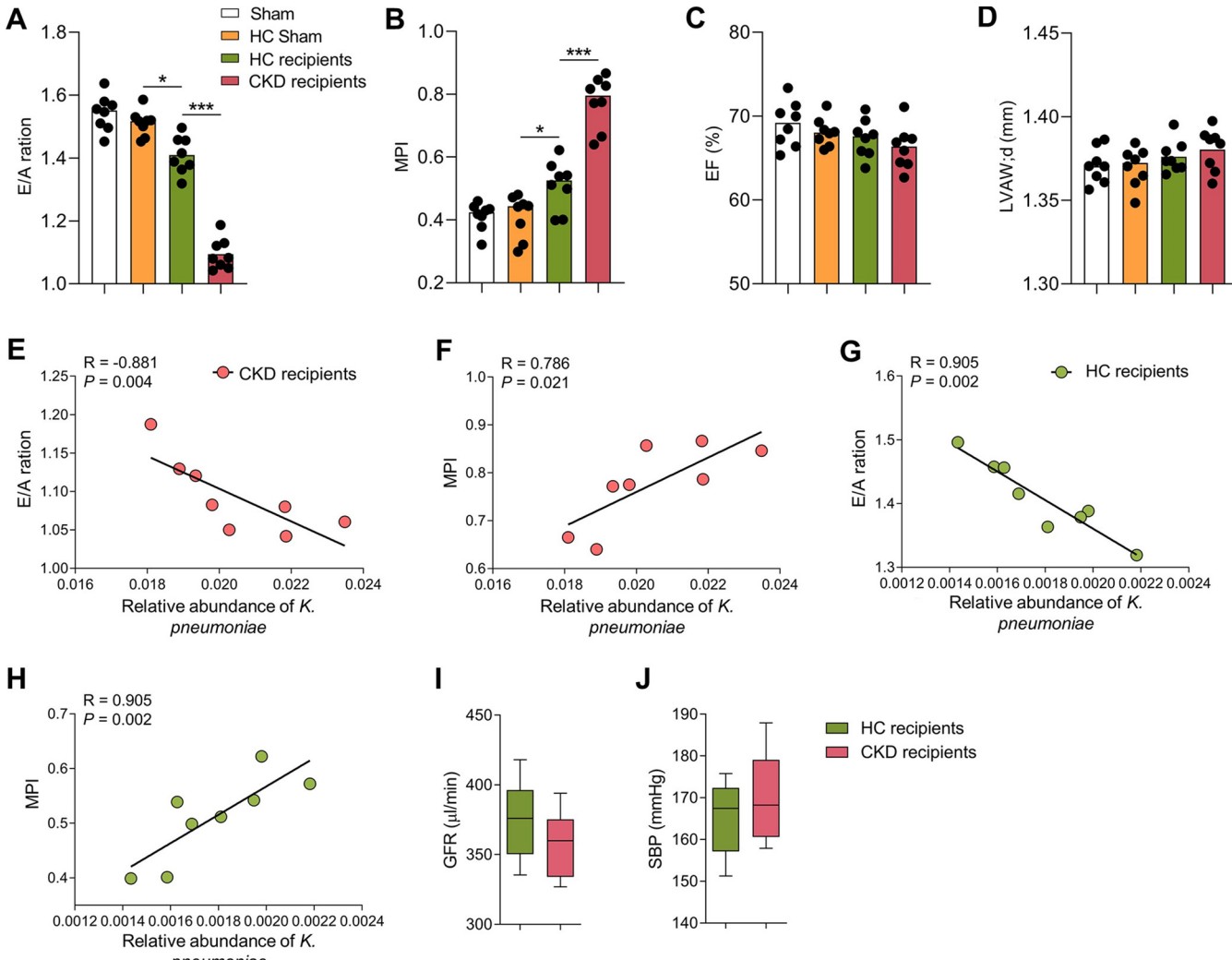

**FIG 3** Gut microbiota from CKD patients exacerbated diastolic dysfunction in the mice. (A-D) Echocardiographic parameters in recipient mice or sham mice with E/A ratio (A), MPI (B), EF (C), and LVAW;d (D) ($n = 8$ per group). (E–H) Correlations between the relative abundance of *K. pneumoniae* and E/A ratio in CKD recipients (E), MPI in CKD recipients (F), E/A ratio in HC recipients (G) and MPI in HC recipients (H) ($n = 8$ per group). (I–J) GFR (I) and SBP (J) in CKD recipients or HC recipients after FMT ($n = 8$ per group). Boxes and whiskers show mean ± interquartile range. *, $P < 0.05$; **, $P < 0.01$; ***, $P < 0.001$.

**K. pneumoniae aggravated cardiac IFNγ⁺ CD4⁺ T cells infiltration and diastolic dysfunction.** To test the effect of Th1-inducing bacteria on UCM, we gavaged *K. pneumoniae* to mouse model of CKD. As control, another group of mice were gavaged with sterile saline (Fig. 5A). Gavage with *K. pneumoniae* enhanced its abundance in stool matter significantly without significant effect on the overall composition of the gut microbiome (Fig. 5B and C). Importantly, mice gavaged with *K. pneumoniae* showed significantly increased cardiac IFNγ⁺ CD4⁺ T cells infiltration compared to the control mice (Fig. 5D), concomitantly with more severe diastolic dysfunction (Fig. 5E and F) and preserved EF and LVAW;d (Fig. 5G and H). Moreover, the relative abundance of *K. pneumoniae* correlated directly with levels of heart-infiltrating IFNγ⁺ CD4⁺ T cells (Fig. 5I). We observed no significant differences between two groups in regarding to SBP or renal function (Fig. 5J and K). Collectively, these findings indicate that *K. pneumoniae* aggravated UCM in the mouse model via the induction of heart-infiltrating IFNγ⁺ CD4⁺ T cells expansion.

**Probiotic decreased abundance of K. pneumoniae and ameliorated diastolic dysfunction.** Given the critical role of *K. pneumoniae* in development of UCM, we point the hypothesis that decrease of the driver species abundance should ameliorate the

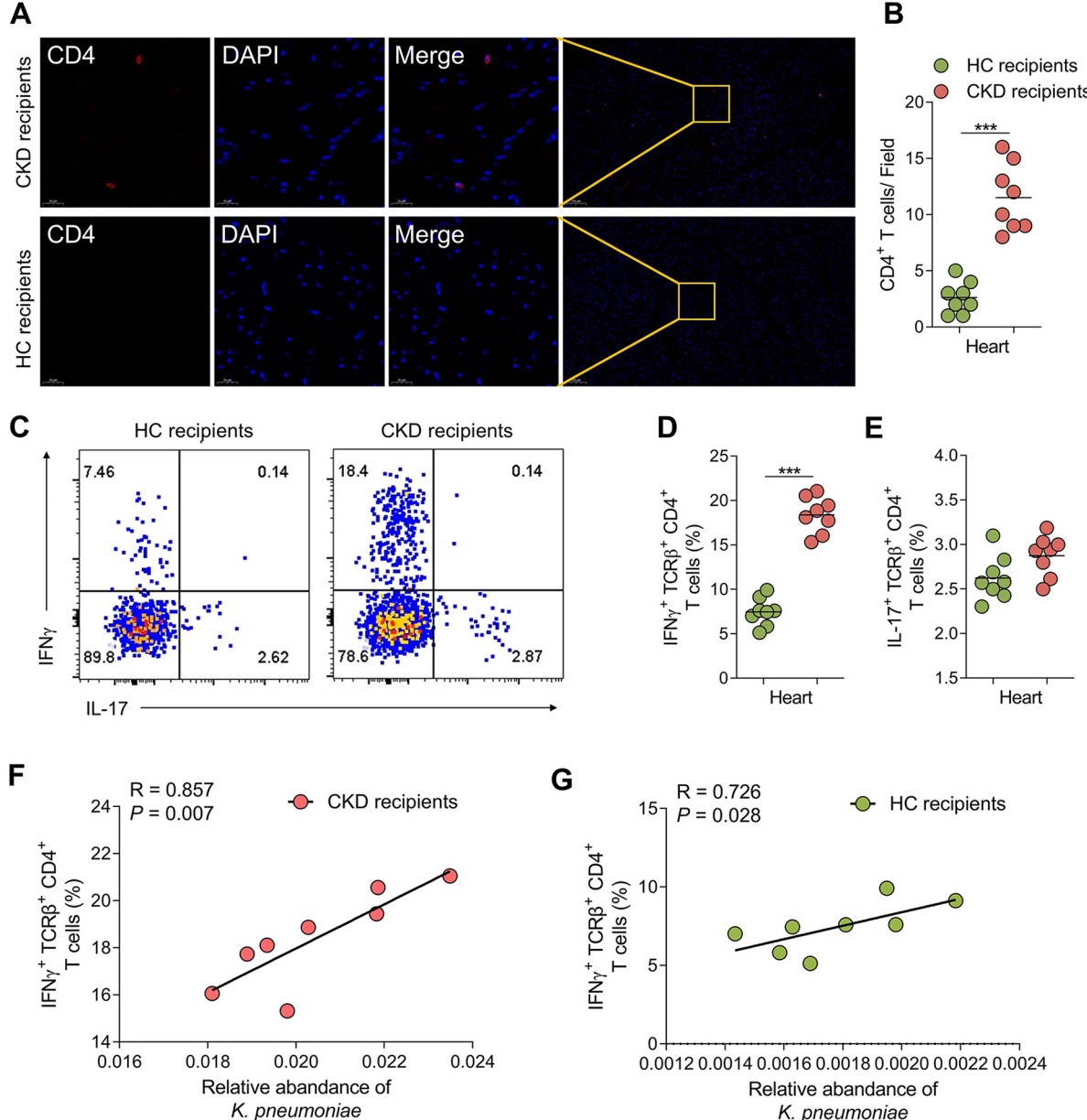

**FIG 4** Gut microbiota from CKD patients induced expansion of heart-infiltrating IFNγ+ CD4+ T cells in the mice. (A) Immunofluorescence staining of CD4+ (red) T cells in cardiac tissue. Nuclei were stained by DAPI (blue). Bar, 20 μm. (B) Quantification of CD4+ T cells in cardiac tissue of HC recipients and CKD recipients (*n* = 8 per group). (C) Flow cytometry-based cytokine profile analysis of heart-infiltrating TCRβ+ CD4+ T cells in CKD recipients or HC recipients (*n* = 8 per group). (D–E) IFNγ+ (D) and IL-17+ (E) producing heart-infiltrating CD4+ T cells from CKD recipients or HC recipients. Dots represent individual mice and lines indicate median values (*n* = 8 per group). (F-G) Correlations between the relative abundance of *K. pneumoniae* and levels of cardiac IFNγ+ TCRβ+ CD4+ T cells in CKD recipients (F) or HC recipients (G) (*n* = 8 per group). *, *P* < 0.05; **, *P* < 0.01; ***, *P* < 0.001.

severity of UCM. Hence, we gavaged *Bifidobacterium animalis* (*B. animalis*), a health-promoting probiotic, to mouse model of CKD (Fig. 6A) and observed that, the abundance of *K. pneumoniae* in the fecal samples was markedly decreased even if the overall microbial composition was not changed significantly (Fig. 6B and C). Notably, there was a significant decrease in cardiac IFNγ+ CD4+ T cells infiltration in mice gavaged with *B. animalis* compared to the control mice (Fig. 6D). In parallel, diastolic dysfunction was significantly improved (Fig. 6E and F). There were no significant differences in terms of EF and LVAW;d between the two groups (Fig. 6G and H). Furthermore, no significant differences of SBP and renal function were observed among the two studied

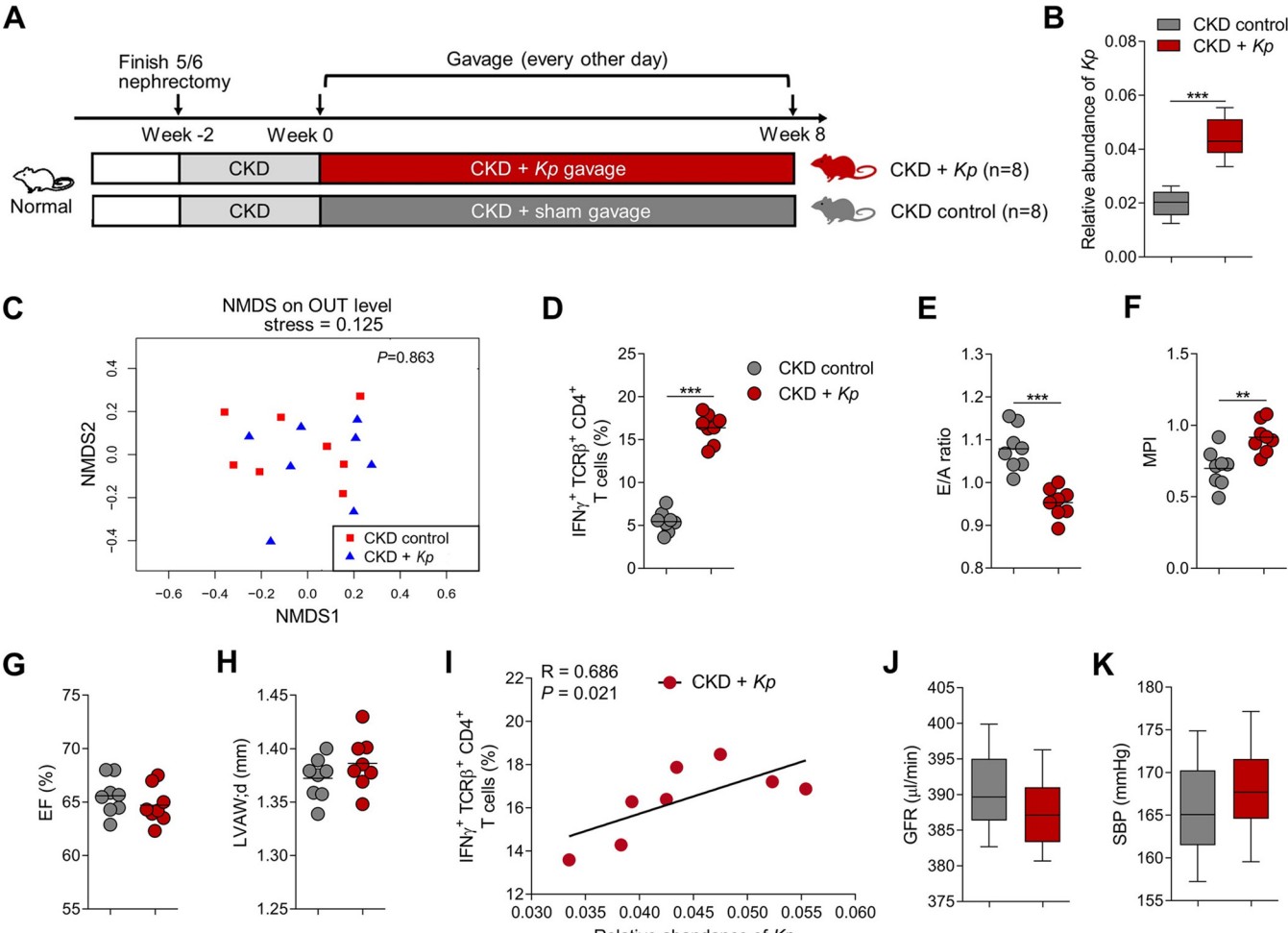

**FIG 5** Effects of *K. pneumoniae* supplementation on cardiac IFNγ⁺ CD4⁺ T cells infiltration and diastolic dysfunction. (A) Experimental design (*n* = 8 per group). (B) The relative abundance of *Kp* in fecal samples of bacterial-gavaged or sham-gavaged mice. Boxes and whiskers show mean ± interquartile range (*n* = 8 per group). (C) NMDS revealed no significant alternations of the gut microbial composition of *Kp*-gavaged mice compared with sham-gavaged mice (*n* = 8 per group). (D) IFNγ⁺ TCRβ⁺ CD4⁺ T cells in the heart tissue of bacterial-gavaged or sham-gavaged mice. Dots represent individual mice and lines indicate median values (*n* = 8 per group). (E–H) Echocardiographic parameters in bacterial-gavaged or sham-gavaged mice with E/A ratio (E), MPI (F), EF (G), and LVAW;d (H). Dots represent individual mice and lines indicate median values (*n* = 8 per group). (I) Correlations between the relative abundance of *Kp* and levels of cardiac IFNγ⁺ TCRβ⁺ CD4⁺ T cells in *Kp*-gavaged mice (*n* = 8 per group). (J–K) GFR (J) and SBP (K) in bacterial-gavaged or sham-gavaged mice. Boxes and whiskers show mean ± interquartile range (*n* = 8 per group). Statistical analysis was performed using Student's *t* test. *, $P < 0.05$; **, $P < 0.01$; ***, $P < 0.001$. *Kp*, *K. pneumoniae*.

groups (Fig. 6I and J). Together, these results suggest that supplementation with probiotic can ameliorate development of UCM in mouse model via modulation of IFNγ⁺ CD4⁺ T cell-inducing bacteria.

## DISCUSSION

UCM is one of the most severe comorbidities of CKD and correlates with increased mortality in patients with CKD (1, 2), yet the pathogenesis of UCM remains incompletely understood, limiting therapeutic approaches (10). The present work aimed to explore these answers in the gut microbiota. We found that the aberrant gut microbiome in CKD patients, particularly *K. pneumoniae*, led to aggravated diastolic dysfunction in mouse model of CKD. One possible mechanism through which this happen might be the abnormal immune responses in extraintestinal tissue induced by the aberrant intestine bacteria, as we observed higher levels of heart-infiltrating IFNγ⁺ CD4⁺ T cells in CKD recipients. Furthermore, we detected that administration of probiotic decreased abundance of *K. pneumoniae* and ameliorated diastolic dysfunction in

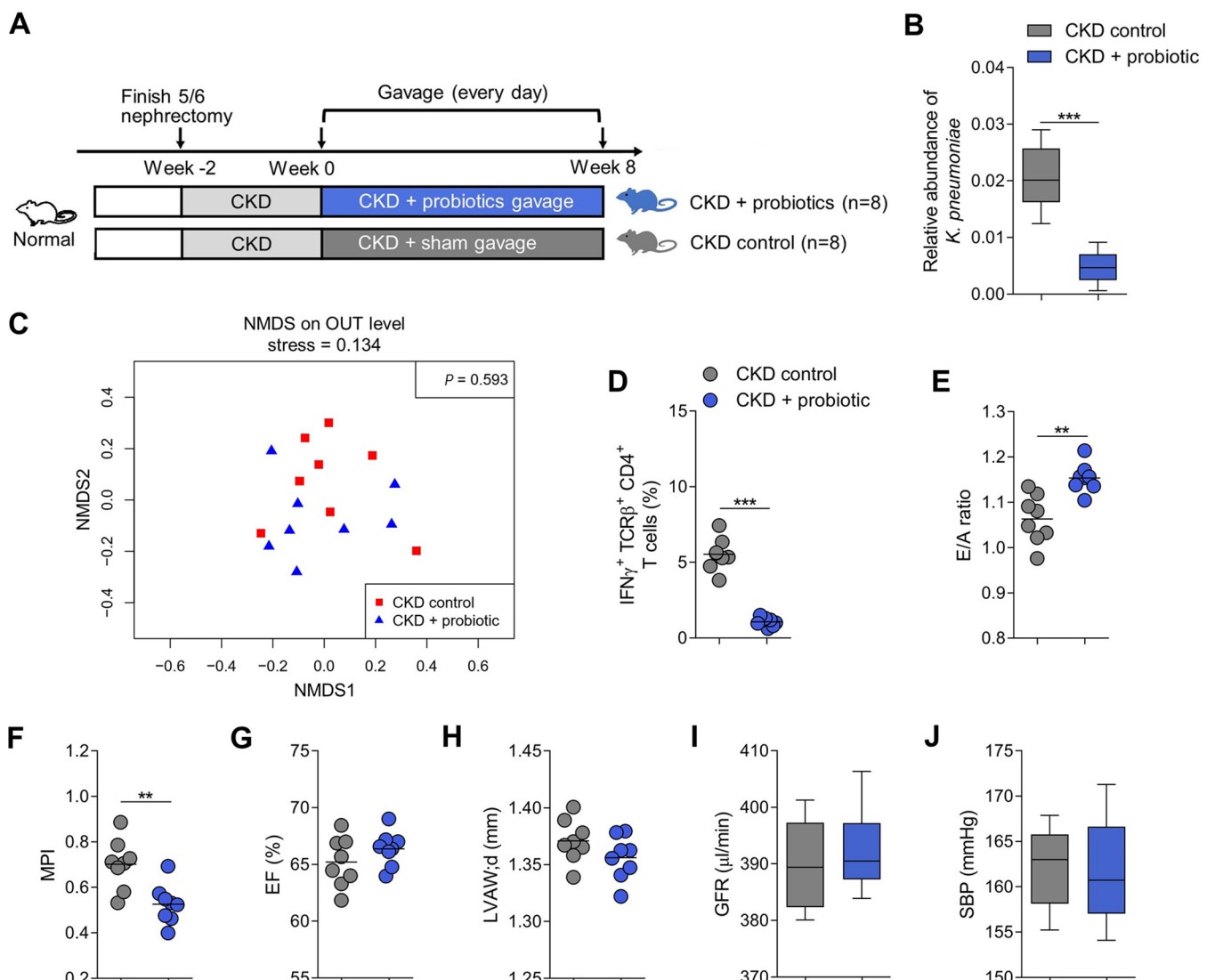

**FIG 6** Effects of probiotic supplementation on relative abundance of *K. pneumoniae*, levels of cardiac IFNγ+ CD4+ T cells, and diastolic dysfunction. (A) Experimental design (*n* = 8 per group). (B) The relative abundance of *K. pneumoniae* in fecal samples of probiotic-gavaged or sham-gavaged mice. Boxes and whiskers show mean ± interquartile range (*n* = 8 per group). (C) NMDS revealed no significant alternations of the gut microbial composition of probiotic-gavaged mice compared with sham-gavaged mice (*n* = 8 per group). (D) IFNγ+ TCRβ+ CD4+ T cells in the heart tissue of bacterial-gavaged or sham-gavaged mice. Dots represent individual mice and lines indicate median values (*n* = 8 per group). (E-H) Echocardiographic parameters in probiotic-gavaged or sham-gavaged mice with E/A ratio (E), MPI (F), EF (G), and LVAW;d (H). Dots represent individual mice and lines indicate median values (*n* = 8 per group). (I–J) GFR (I) and SBP (J) in probiotic-gavaged or sham-gavaged mice. Boxes and whiskers show mean ± interquartile range (*n* = 8 per group). Statistical analysis was performed using Student's *t* test. *, *P* < 0.05; **, *P* < 0.01; ***, *P* < 0.001.

the mice. This study proposed that a Gut Microbiota-Gut-Kidney-Heart axis could play a critical role in elucidating the etiology of UCM.

In the current study, we found that patients with CKD exhibited reduced bacterial diversity with a severe gut dysbiosis, consistent with the findings of previous reports (26, 27). Specially, *K. pneumoniae*, which is believed to be a strong inducer of IFNγ+ CD4+ T cells, was enriched in patients with CKD. Furthermore, using FMT and administration of certain gut commensals, we established a causative role of the abnormal gut microbiota, particularly *K. pneumoniae*, in favoring development of diastolic dysfunction during UCM in the mice, which is supported by a clinical study that suggested an association between changes in gut microbiota and diastolic dysfunction in population with type 2 diabetes mellitus (31).

Diastolic dysfunction that precedes the development of left ventricular hypertrophy produces severe symptoms of heart failure and is an independent predictor for cardiovascular

death among in individuals with CKD (32–35). Similarly, diastolic dysfunction has been proven to precede the development of left ventricular hypertrophy and elevated BP during essential hypertension (36, 37). Therefore, diastolic dysfunction should be considered clinically adverse outcomes in preclinical and translational studies of UCM.

Gut microbiota plays a critical role in the modulation of host immunity and disease (21–23). Dysbiosis of gut microbiome is involved in the pathogenesis of certain extraintestinal disease through the induction of T cell immune responses (38–41). In the present study, we found that the CKD-associated gut microbiota, particularly *K. pneumoniae*, induced the expansion of IFNγ⁺ CD4⁺ T cells in the heart in the mouse model. Enrichment of *K. pneumoniae* in CKD is especially worth noting as it is a newly discovered pathogen involved in inducing nonalcoholic fatty liver disease via acting on immune cells (42). Hence, *K. pneumoniae* could be the critical regulator of the Gut Microbiota-Gut-Kidney-Heart axis in UCM.

Accumulation of T cells has been described in patients with CKD and associated with cardiovascular events in this patient population (19, 43, 44). Recently, a plethora of studies have shown a pathogenic role for T cells in cardiac dysfunction and cardiac remodeling during chronic heart failure (14–18). An animal study of CKD by Winterberg et al. identified that T cells, especially IFNγ⁺ CD4⁺ T cells, infiltrated the heart and led to diastolic dysfunction, whereas depletion of T cells significantly ameliorated diastolic function without altering left ventricular hypertrophy, hypertension or renal dysfunction, suggesting a causal role for T cells in diastolic dysfunction during UCM (20). In this work, we found that the abnormal gut microbiota from patients with CKD, especially *K. pneumoniae*, induced expansion of heart-infiltrating IFNγ⁺ CD4⁺ T cells and exacerbated diastolic dysfunction in the mice. Additionally, the abundance of *K. pneumoniae* correlated strongly with levels of heart-infiltrating IFNγ⁺ CD4⁺ T cells in the mouse model. Collectively, these results support the notion that the gut microbiota, particularly *K. pneumoniae*, derived UCM at least in part via the induction of cardiac IFNγ⁺ CD4⁺ T cells expansion.

Microbial interventions, e.g., FMT, probiotics, prebiotics, and antibiotics, may be effective tools to treat or prevent acute and chronic diseases (45). These interventions have notable advantages such as low cost production and effectiveness, but are accompanied by disadvantages such as diarrhea, induction of resistance, and transmission of microbiota-associated diseases (46). Current treatments, such as antihypertensive or lipid-lowering treatment, improve clinical prognosis in UCM only modestly at best. Emerging study revealed that manipulation of the gut microbiota improved cardiac function in mice with heart failure (25). In the present study, we detected that administration of probiotic decreased abundance of *K. pneumoniae* and ameliorated diastolic dysfunction in the mice. Hence, there is a major need in future clinical trials of patients with CKD to explore if modulation of gut microbiota by FMT, probiotic or antibiotics can alleviate cardiac dysfunction, reduce mortality, and improve life quality.

Dialysis is the primary modality of renal replacement therapy for patients with end-stage renal disease. However, more frequent dialysis may increase the risk of sepsis, which induces cardiac inflammation and function alteration (47). Moreover, dialysis has been associated with development of cardiomyopathy in this patient population (48). Manipulation of gut microbiota by FMT has been observed to significantly reduced cardiac inflammation and interstitial fibrosis and improve cardiac function in mice with DOX-induced cardiomyopathy (49). In the present study, we found that administration of probiotic ameliorated diastolic dysfunction in the mice via decreasing abundance of intestinal *K. pneumoniae* and reducing levels of cardiac IFNγ⁺ CD4⁺ T cells.

There are several limitations in this study. First, all individuals were of Chinese descent only, which limited generalization of the current results. Second, we cannot exclude the contribution of administration of antibiotics to the immune phenomena observed in the present study because broad-spectrum antibiotics treatment has been shown to exert an immunosuppressive effect on immune cell homeostasis in the gut (50, 51). Lastly, even though we found that IFNγ⁺ CD4⁺ T cells proliferated initially in

the gut and they subsequently migrated to the cardiac tissue in the mice, the exact pathway through which IFN$\gamma^+$ CD4$^+$ T cells completes its migration also needs further research.

In conclusion, despite the limitations mentioned above, this study demonstrates that aberrant gut microbiota in patients with CKD, especially *K. pneumoniae*, contributes to the pathogenesis of UCM through the induction of heart-infiltrating IFN$\gamma^+$ CD4$^+$ T cells expansion, suggesting that manipulation of the gut microbiome may serve as a promising target for the amelioration of UCM.

## MATERIALS AND METHODS

**Study cohort and sample collection.** We performed a case-control study to compare the gut microbiota of patients with CKD and healthy controls by 16S rRNA (rRNA) gene sequencing. Patients with CKD were consecutively recruited from the Nephrology Center of Shanghai General Hospital Affiliated to Shanghai Jiaotong University School of Medicine in Shanghai, China. All participants were diagnosed with CKD according *to Kidney Disease: Improving Global Outcomes Clinical Practice* guidelines. Exclusion criteria of the human study were as follows: a) CKD-related drug treatment or dialysis had been initiated; b) antibiotics or probiotic therapy in the past 4 weeks; c) history of other diseases such as digestive disease, liver disease, diabetes, and cancer; and d) patients missing clinical data. Meanwhile, we recruited sex-, age-, and BMI-matched healthy volunteers from the physical examination center of the hospital. Exclusion criteria for healthy controls included diabetes, hypertension, dyslipidemia, obesity, digestive disease, liver disease, abnormal kidney function, and cancer. Individuals were excluded if they had taken probiotic or antibiotics within 4 weeks. The study protocol adhering to the Declaration of Helsinki was approved by the Ethics Committees of Shanghai General Hospital. The written informed consents were obtained from the participants. Demographics and clinical data of participants were collected from electronic medical records of Shanghai General Hospital. Fresh stool samples were collected from all participants between 06:00 and 08:00 a.m. and stored at −80℃ immediately.

**16S rRNA gene sequencing.** Total stool DNA was extracted using the QIAamp Fast DNA Stool minikit (51604, Qiagen, Hilden, Germany). DNA samples were sent to Majorbio Bio-pharm Technology (Shanghai, China) for analysis of microbial composition and quantification of the relative abundance of each member by 16S rRNA high-throughput gene sequencing. The extracted DNA was quality-checked by agarose gel electrophoresis and quantified using NanoDrop 2000 (Thermo Scientific, USA). Primers targeting hypervariable V3–V4 region (338F/806R) of 16S rRNA gene were performed to amplify extracted DNA sample by PCR. The amplicons were purified by AxyPrep DNA Gel (Axygen, CA, USA) and quantitatively analyzed using Qubit2·0 (Invitrogen, USA). The amplicon sequencing was conducted by Majorbio Bio-pharm Technology Co., Ltd., on the Illumina MiSeq platform, based on a paired-end sequencing mode (250 bp, V3–V4).

**Bioinformatics analysis.** Raw 16S rRNA gene amplicon sequencing reads were subjected to quality control by UPARSE filtering of low-quality and chimeric sequences (52). Qualified reads were clustered into OTUs at 97% similarity threshold by Usearch (53). Microbiome diversity was displayed by the Shannon index and Chao1 index using R program package "vegan" for calculations (54). Principal-component analysis (PCA), principal coordinate analysis (PCoA), and nonmetric multidimensional scaling (NMDS) were performed to differentiate the microbiota composition between groups using the R package. Microbial taxonomic comparison between groups was test by the Wilcoxon rank-sum test.

**Animal experiments. (i) Experiment mice.** All experiments described here were approved by the Institutional Animal Care and Use Committee (IACUC) of Shanghai General Hospital (2021AW021) and were conducted strictly in accordance with the approved animal study protocol. The mice used in present study are 129 × 1/SvJ littermates (male, 5 weeks) purchased from Jackson Laboratories. CKD model was induced in the mice through 5/6th nephrectomy (35), which reliably replicates the clinical characteristics of UCM (55, 56). The mice were housed in the Laboratory Animal Center of Shanghai General Hospital and allowed to acclimatize for 2 weeks before the experiment. All mice were bred and maintained in a specific-pathogen-free (SPF) facility with the same 12 h light-dark cycle, food, and water. Administration of experimental compounds to the mice was achieved by using the gavage technique. Gavage is the introduction of a compound into animal's stomach using a special feeding needle (57). All mice with gavage treatment showed no stress, inflammations, or tissue damage.

**(ii) Collection of fecal samples from donors.** Ten CKD individuals and 10 healthy volunteers from the original subjects were selected as donors by the following criteria: (i) had the willingness to come for a hospital visit; (ii) had the highest levels of control-enriched and CKD-enriched species, respectively. Fresh stool samples were collected from donors in the same group and were stored at 4℃. Approximately 2.5 g of stool matter was diluted by 25 mL of sterile saline. The suspensions were allowed to stand under gravity for 5 min, and supernatants were frozen at −80℃ with sterile glycerol. Frozen stool supernatants were thawed in equivalent volume before FMT.

**(iii) Fecal microbiota transplantation experiment.** Two weeks after 5/6 nephrectomy surgery or sham operation, levels of serum urea and creatinine were markedly higher in nephrectomized rats than in sham-operated rats, indicating CKD model was successfully induced. Depletion of gut microbiome before FMT was achieved by administering mice an antibiotic cocktail (0.5 g/l vancomycin, 1 g/l metronidazole, 1 g/l neomycin, and 1 g/l ampicillin) via daily gavage for 3 weeks (58, 59). CKD rats were then randomly divided into two groups as recipient mice transplanted with stool samples from CKD patients (CKD recipients) and the recipient mice receiving stool samples from HC recipients. The recipient mice

were transplanted with stool microbiota suspension (1 mL/mice) by daily gavage for the following 3 weeks (Fig. 2A).

**(iv) Bacterial strains and culture conditions.** *K. pneumoniae* (BNCC102997 = ATCC10031) and *B. animalis* (BNCC186305 = ATCC25527) were purchased from Beijing BeiNa Biotechnology Institute. *K. pneumoniae* was grown overnight aerobically at 37°C in Brain Heart Medium (Becton, Dickinson). *B. animalis* was cultured anaerobically overnight in MRS broth (AOBOX) at 37°C. Bacterial cells were centrifuged and resuspended in PBS at level of $1 \times 10^9$ colony forming units (CFU)/mL for *K. pneumoniae* and $4 \times 10^{10}$ CFU/mL for *B. animalis*, respectively.

**(v) Single strain gavage experiments.** Two independent animal experiments were conducted to test effect of *K. pneumoniae* and probiotic on UCM. (1) *K. pneumoniae gavage experiment*. 2 weeks after surgery, the mice were randomly assigned to two groups as CKD + *K. pneumoniae* rats and CKD + control rats. In the following 8 weeks, the rats were gavaged every other day with 1 mL of *K. pneumoniae* or sterile saline (Fi. 5A). (2) *Probiotic gavage experiment*. 2 weeks post operation, the mice were randomly divided into two groups as CKD + probiotic mice and CKD + control mice. In the following 8 weeks, the mice were gavaged daily with 1 mL of *B. animalis* or sterile saline (Fig. 6A).

**(vi) Sample collection.** Fresh stool samples were obtained from model mice and immediately stored at −80°C for 16S rRNA gene sequencing. Tissue samples were isolated for single-cell suspensions generation and immunofluorescence. Serum was obtained and immediately stored at −80°C.

**Echocardiography.** Transthoracic echocardiography was conducted on mice under 2% isoflurane anesthesia using high-resolution ultrasound system (Vevo 2100, Visualsonics). Diastolic function was determined by measuring the ratio of the mitral annulus early and late peak flow velocities (E/A) and by calculating myocardial performance index (MPI) (20). Systolic function was evaluated by measuring left ventricular ejection fraction (EF) (20). All measurements were conducted by a technician who was blinded to assignments of model mice.

**Immunofluorescence.** Heart tissues were frozen in OCT tissue embedding compound at −80°C. Cryosections of 5 $\mu$m were blocked in PBS with 2% goat serum (Sigma-Aldrich), 1% BSA, and 1 mg/mL anti-Fc$\gamma$ receptor (BD Biosciences). Sections were incubated overnight at 4°C with anti-mouse CD4 (BD Bioscience). Nuclei were stained by DAPI.

**Generation of mononuclear cells.** Hearts were perfused with PBS and then cut into pieces and incubated in RPMI 1640 containing 10% FBS (Lonza), 1 mg/mL collagenase B (Sigma), and 25 $\mu$g/mL DNase I (Applichem) at 37°C under constant agitation. Supernatant was then collected in 10 mL of RPMI 1640 with 10% FBS. All mononuclear cells were further purified by centrifugation (20 min at $800 \times g$) at 4°C on 30%–70% Percoll gradient (GE Healthcare).

**Flow cytometry.** Flow cytometry was conducted on a BD LSRFortessa (BD Biosciences) and the data were further analyzed using FlowJo software (Treestar Inc.). Cells were stimulated in RPMI 1640 with 10% FBS (Lonza) and 2 $\mu$L of leukocyte activation cocktail With GolgiPlug (BD Bioscience) for 6 h at 37°C. The live cells were discriminated by Fixable Viability Stain 780 (Biolegend). Cell surfaces were stained at 4°C for 30 min with appropriate fluorescence-labeled antibodies against CD3, CD4, and TCR$\beta$ (BD Biosciences). After the fixation and permeabilization using cytofix-cytoperm (BD Biosciences), intracellular cytokines were stained using appropriate fluorescence-labeled antibodies to interferon $\gamma$ (IFN-$\gamma$) and interleukin-17A (IL-17A) (BD Biosciences) for 30 min at 4°C.

**Statistical analysis.** Data are shown as mean ± standard deviation (SD), percent frequency, or median (interquartile range [IQR]). Comparison between the groups was performed using Student's *t* test or Mann-Whitney *U* test for continuous variables and Pearson's $\chi^2$ test for categorical variables. Multiple comparisons were conducted by One-way ANOVA with Tukey's posttest or Kruskall-Wallis *H* test for nonparametric variables. Statistical analyses were carried out using SPSS 22.0 (Chicago, IL, USA). A two-tailed $P < 0.05$ were considered statistically significant.

**Data availability.** The raw sequencing data have been deposited to the NCBI Sequence Read Archive (SRA) database under accession number PRJNA847979.

## ACKNOWLEDGMENTS

We have no conflicts of interest to declare.

This work was supported by the National Natural Science Foundation of China (81970624).

W.Y., B.H., and X.Z. designed the study. L.W., X.Z., and B.H. performed the experiments and analyzed the data. B.H. and X.Z. wrote the manuscript. W.Y. supervised the experiments and reviewed the manuscript. All authors read and approved the final manuscript.

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
