## [Reviewer comments · Microbiology Spectrum]

Microbiology Spectrum

Dysbiosis of Gut Microbiota Contributes to Uremic Cardiomyopathy via Induction of IFN γ -producing CD4⁺ T Cells Expansion

Bin Han, Xiaoqian Zhang, Ling Wang, and Weijie Yuan

Corresponding Author(s): Weijie Yuan, Shanghai General Hospital

Review Timeline:

Submission Date:	August 12, 2022
Editorial Decision:	September 30, 2022
Revision Received:	October 19, 2022
Editorial Decision:	November 10, 2022
Revision Received:	November 11, 2022
Accepted:	November 18, 2022

Editor: Pei-Yuan Qian

Reviewer(s): The reviewers have opted to remain anonymous.

Transaction Report:

DOI: <https://doi.org/10.1128/spectrum.03101-22>

September 30, 2022

Prof. Weijie Yuan
Shanghai General Hospital
85 Wujin Road
Shanghai
China

Re: Spectrum03101-22 (**Dysbiosis of Gut Microbiota Contributes to Uremic Cardiomyopathy via Induction of Th1 Cells Expansion**)

Dear Prof. Weijie Yuan:

Dear Authors,

While reviewers recognized the merits of your work, they raised a number of issues/concerns that need to be carefully addressed before this ms can be further considered. Please go through the comments and response accordingly (point-to-point) in your revision. Please submit with your revised ms (one clean copy and one tracked copy), together with your response.

Best,

Pei-Yuan Qian

Link Not Available

Sincerely,

Pei-Yuan Qian

Journals Department
Reviewer comments:

Reviewer #1 (Comments for the Author):

This is an important report shedding some light on the innovative treatment against the complications of chronic kidney disease, such as uremic cardiopathy, which may be prevented by intensive and regular dialysis treatment to better maintain renal function. However, more frequent dialysis may increase the risk of sepsis in patients. Therefore, the manipulation of the gut microflora by probiotics, fecal microbiota transplant and antibiotic as the treatment of uremic cardiopathy symptoms, such as hypertension, aortic stenosis resulting in left ventricular hypertrophy, is a plausible idea.

Major comments:

- 1) The rationale of this study should be clear stated, i.e., why the advantage of manipulation of gut microbiota over the prevention of uremic cardiomyopathy in chronic kidney disease patients by the kidney dialysis.
- 2) In Abstract, specify the mouse model of chronic kidney disease.
- 3) It would be useful to state the Th1 cell population with specific T cell specific markers throughout text.
- 4) Authors may consider the GM-Gut-Kidney-Heart axis instead of just Gut-heart.
- 5) Consider acute vs chronic therapy using fecal microbiota transplantation, prebiotics, probiotics and/or antibodies and discuss the potential benefits and disadvantages.
- 6) Please add comments on BMI and status of diabetes and the severity of renal and cardiac complications. It is interesting that there is no difference in BMI, although almost 30% of patients are diabetic.
- 7) Did animals with gavage treatment show any stress, inflammations and tissue damage, which may alter the gut microflora?
- 8) In Figure 4A, it would be useful to also include images of heart sections after control treatment. Also, authors should show a low magnification micrograph to point out the sampled heart region used for sections.

Minor comments:

- 1) Follow the proper format for referencing the previous report. See an example on the line 81 and 82.
- 2) Please consider restructuring of some sentences, such as line 26, "in UCM pathogenesis" should be "in the pathogenesis of UCM", and etc.
- 3) Define abbreviations when they appear for the first time and then use them when they appear in subsequent text. The abbreviation table can be included so that there is no need to define abbreviation in every figure legend and text, again.
- 4) Define the abbreviation of bacteria, such as *K. pneumoniae* for *Klebsiella pneumoniae*, and then continue to use it throughout text. Authors sometimes used *K. pneumoniae* or *Klebsiella* or *pneumoniae*.
- 6) Be consistent in describing the *Bifidobacterium animalis*, such as *B. animalis* throughout text.
- 7) Correct typos and grammatical mistakes, such as line 351, add comma after *animalis* before respectively.
- 8) Please check line 703. I do not know the meaning of "whiskers" in this context?
- 9) Define probiotic-gavage and bacterial-gavage in the method and use them throughout text. I assume that they are same experimental group.

Reviewer #2 (Comments for the Author):

This manuscript describes the pathogenic role for cardiac Th1 cells in Uremic Cardiomyopathy and the elevated abundance of opportunistic pathogen *Klebsiella pneumoniae* in CKD, and role of gut microbiota (*K. pneumoniae* in particular), and its contribution to the development of UCM via the induction of heart-infiltrating Th1 cells expansion. According to the authors' findings, they proposed that manipulation of the gut microbiome might serve as a promising target for the amelioration of UCM. Overall, this study is interesting and important for the UCM study and therapy. The manuscript is well organized and the data interpretation is convincing. However, some specific comments should be addressed.

Specific comments:

- L28: Please insert "gene" before "sequencing".
- L80: "...is strongly correlated to..."
- L81-82: Please delete "Journal of American Society of Nephrology".
- L87: Gut microbiome is highly dynamic. Changes of gut microbiome can cause gut dysbiosis but is NOT termed gut dysbiosis. The authors should revise this statement carefully.
- L91: "Emerging studies have..."
- L92-93: The Enterobacteriaceae is a large family of bacteria, including many familiar pathogens, such as the genera *Salmonella*, *Shigella* and *Escherichia*. *Klebsiella* is also a genus of it. Which members of Enterobacteriaceae have been changed? The authors should make it clear.
- L93: Please revise "*Klebsiella* are" to "Members of *Klebsiella* are"
- L106-108: As shown in Figure 1A, the significantly decreased microbial diversity of CKD group compared to the control should be stated here.
- L113-121: The purpose of this part is to show the experimental design and validity of FMT technique used?
- L123-125: Repeat of L113-114.
- L125-127: Figures 3C and 3D showed no significant differences between the four groups studied. Please check it.
- L127-130: The presentation for Figures 3E-3H is problematic. For correlations, no differences between CKD and HC recipients, 3E vs 3G (both negative correlations), 3F vs 3H (both positive correlations). Please check it.

L146-148: Except *K. pneumoniae*, any other microbial taxa showed similar correlation patterns?
L152: "To test ..."
L152-154: Please revise this part to make better sense of statement.
L158: Figure 5C, what's the pattern if PcoA used?
L161: Figures 5G and 5H showed no significant differences between the two groups.
L174: For Figure 6C, please see the comment of Figure 5C.
L177: Figures 6G and 6H showed no significant differences between the two groups.
L178: "...two studied groups..."
L204: "...by a clinical study..."
L255-257: "we believe that administration of antibiotic did not affect the observed differences in immune". This statement doesn't support your current hypothesis and test. Antibiotics do affect immune because they can change gut microbiota largely. Please modify this statement carefully.
L270: "...16S ribosomal RNA (rRNA) gene sequencing."
L302: "...based on a paired-end sequencing mode"
L304: Duplicate sequences should be removed for shotgun sequencing but NOT for amplicon sequencing.
L303-312: Please insert citations for these tools.
L344-351: Please use abbreviations for "*Bifidobacterium animalis*". And other places in the manuscript.
L377: "...stain incubated...?"
L411: "National Natural Science Foundation of China"

Staff Comments:

Preparing Revision Guidelines

Please return the manuscript within 60 days; if you cannot complete the modification within this time period, please contact me. If you do not wish to modify the manuscript and prefer to submit it to another journal, please notify me of your decision immediately so that the manuscript may be formally withdrawn from consideration by Microbiology Spectrum.

This manuscript describes the pathogenic role for cardiac Th1 cells in Uremic Cardiomyopathy and the elevated abundance of opportunistic pathogen *Klebsiella pneumoniae* in CKD, and role of gut microbiota (*K. pneumoniae* in particular), and its contribution to the development of UCM via the induction of heart-infiltrating Th1 cells expansion. According to the authors' findings, they proposed that manipulation of the gut microbiome might serve as a promising target for the amelioration of UCM. Overall, this study is interesting and important for the UCM study and therapy. The manuscript is well organized and the data interpretation is convincing. However, some specific comments should be addressed.

Specific comments:

L28: Please insert "gene" before "sequencing".

L80: "...is strongly correlated to..."

L81-82: Please delete "Journal of American Society of Nephrology".

L87: Gut microbiome is highly dynamic. Changes of gut microbiome can cause gut dysbiosis but is NOT termed gut dysbiosis. The authors should revise this statement carefully.

L91: "Emerging studies have..."

L92-93: The Enterobacteriaceae is a large family of bacteria, including many familiar pathogens, such as the genera *Salmonella*, *Shigella* and *Escherichia*. *Klebsiella* is also a genus of it. Which members of Enterobacteriaceae have been changed? The authors should make it clear.

L93: Please revise "*Klebsiella* are" to "Members of *Klebsiella* are"

L106-108: As shown in Figure 1A, the significantly decreased microbial diversity of CKD group compared to the control should be stated here.

L113-121: The purpose of this part is to show the experimental design and validity of FMT technique used?

L123-125: Repeat of L113-114.

L125-127: Figures 3C and 3D showed no significant differences between the four groups studied. Please check it.

L127-130: The presentation for Figures 3E-3H is problematic. For correlations, no differences between CKD and HC recipients, 3E vs 3G (both negative correlations), 3F vs 3H (both positive correlations). Please check it.

L146-148: Except *K. pneumoniae*, any other microbial taxa showed similar correlation patterns?

L152: "To test ..."

L152-154: Please revise this part to make better sense of statement.

L158: Figure 5C, what's the pattern if PcoA used?

L161: Figures 5G and 5H showed no significant differences between the two groups.

L174: For Figure 6C, please see the comment of Figure 5C.

L177: Figures 6G and 6H showed no significant differences between the two groups.

L178: "...two studied groups..."

L204: "...by a clinical study..."

L255-257: "we believe that administration of antibiotic did not affect the observed differences in immune". This statement doesn't support your current hypothesis and test. Antibiotics do affect immune because they can change gut microbiota largely. Please modify this statement carefully.

L270: "...16S ribosomal RNA (rRNA) gene sequencing."

L302: "...based on a paired-end sequencing mode"

L304: Duplicate sequences should be removed for shotgun sequencing but NOT for amplicon sequencing.

L303-312: Please insert citations for these tools.

L344-351: Please use abbreviations for "Bifidobacterium animalis". And other places in the manuscript.

L377: ...stain incubated...?

L411: "National Natural Science Foundation of China"

Response to Reviewers

Reviewer #1 (Comments for the Author):

This is an important report shedding some light on the innovative treatment against the complications of chronic kidney disease, such as uremic cardiopathy, which may be prevented by intensive and regular dialysis treatment to better maintain renal function. However, more frequent dialysis may increase the risk of sepsis in patients. Therefore, the manipulation of the gut microflora by probiotics, fecal microbiota transplant and antibiotic as the treatment of uremic cardiopathy symptoms, such as hypertension, aortic stenosis resulting in left ventricular hypertrophy, is a plausible idea.

Major comments:

1) The rationale of this study should be clear stated, i.e., why the advantage of manipulation of gut microbiota over the prevention of uremic cardiomyopathy in chronic kidney disease patients by the kidney dialysis.

Answer: Thanks for your careful review and insightful comments. Dialysis is the primary modality of renal replacement therapy for patients with end-stage renal disease. However, more frequent dialysis may increase the risk of sepsis, which induces cardiac inflammation and function alteration (PMID: 36108388). Moreover, dialysis has been associated with development of cardiomyopathy in this patient population (PMID: 24738144). Manipulation of gut microbiota by FMT has been observed to significantly reduced cardiac inflammation and interstitial fibrosis and improve cardiac function in mice with DOX-induced

cardiomyopathy (PMID: 34416387). In the present study, we found that administration of probiotic ameliorated diastolic dysfunction in the mice via decreasing abundance of intestinal *K. pneumoniae* and reducing levels of cardiac IFN γ ⁺ CD4⁺ T Cells. These statements mentioned above have been added to the discussion section of the revised manuscript (Line 271-281, 637-646 in the tracked copy).

2) In Abstract, specify the mouse model of chronic kidney disease.

Answer: Thanks for your constructive comment. In our study, we modeled this condition by inducing chronic kidney disease via 5/6th nephrectomy in mice. We have specified the mouse model of chronic kidney disease in the abstract in the revised manuscript (Line 31 in the tracked copy).

3) It would be useful to state the Th1 cell population with specific T cell specific markers throughout text.

Answer: Thank you very much for raising this important issue. Following your advice, Th1 cells have been amended to IFN γ ⁺ CD4⁺ T Cells throughout text (Line 2, 36-37, 39, 43, 57, 83, 98, 104, 144-145, 147, 152-154, 156, 159, 161-162, 171, 175, 178, 188, 196, 207, 216, 235, 245, 251, 253-254, 256, 290, 292, 297, 729, 744-745, 768 in the tracked copy).

4) Authors may consider the GM-Gut-Kidney-Heart axis instead of just Gut-heart.

Answer: Thanks a lot for your careful review and insightful suggestion. Following your suggestion, we have amended the gut-heart axis to the

GM-Gut-Kidney-Heart axis in the revised manuscript (Line 43-44, 50-51, 211-212, 239 in the tracked copy).

5) Consider acute vs chronic therapy using fecal microbiota transplantation, prebiotics, probiotics and/or antibodies and discuss the potential benefits and disadvantages.

Answer: Thanks for your careful review and critical comments. Microbial interventions, e.g., FMT, probiotic, prebiotics, and antibiotics, may be effective tools to treat or prevent acute and chronic diseases (PMID: 35333590). These interventions have notable advantages such as low cost production and effectiveness, but are accompanied by disadvantages such as diarrhea, induction of resistance, and transmission of microbiota-associated diseases (PMID: 32114770). These statements mentioned above have been added to the discussion section of the revised manuscript (Line 257-262, 628-636 in the tracked copy).

6) Please add comments on BMI and status of diabetes and the severity of renal and cardiac complications. It is interesting that there is no difference in BMI, although almost 30% of patients are diabetic.

Answer: Thank you for carefully and patiently reviewing our manuscript. Among 122 patients with CKD in this study, 36 (29.5%) had a current diabetes and their mean BMI was $22.0 \pm 4.82 \text{ kg/m}^2$, which has been added to the results section in the revised manuscript (Line 110-111 in the tracked copy). In fact, we recruited sex-, age-, and BMI-matched healthy volunteers from the physical examination center of the hospital (Line in the tracked copy 109-110, 313, 683). Hence, there was no difference in BMI between the two studied groups.

Among patients, the proportions were 28.7% for CKD stage 3, 33.6% for stage 4, and 37.7% for stage 5, which has been added to Table 1 in the revised manuscript (Line 683 in the tracked copy). Unfortunately, we cannot portray the severity of cardiac complications in patients because majority of the patients did not undergo the cardiac structural and functional examination (e.g., echocardiography). We will take your valuable recommendation in mind and would perform it in our future study.

7) Did animals with gavage treatment show any stress, inflammations, and tissue damage, which may alter the gut microflora?

Answer: Thanks for your careful review. We are very sorry for missing the important information. In our study, all mice with gavage treatment showed no stress, inflammations, or tissue damage. We have added the important information to the methods section of the revised manuscript (Line 361-362 in the tracked copy).

8) In Figure 4A, it would be useful to also include images of heart sections after control treatment. Also, authors should show a low magnification micrograph to point out the sampled heart region used for sections.

Answer: Thanks for your careful review and insightful comments. Following your advice, we have added images of heart sections after control treatment to Figure 4A in the revised manuscript (Revised Figure 4A). Additionally, we have shown a low magnification micrograph to point out the sampled heart region used for sections to Figure 4A in the revised manuscript (Revised Figure 4A).

Minor comments:

1) Follow the proper format for referencing the previous report. See an example on the line 81 and 82.

Answer: Thanks for your careful review. Following your advice, we have improved the format for referencing the previous report in our manuscript. For instance, “In a mouse model of CKD, Winterberg et al. found that T cells.....” has been amended to “An animal study of CKD by Winterberg et al. identified that T cells.....” in the revised manuscript (Line 244-245 in the tracked copy).

2) Please consider restructuring of some sentences, such as line 26, "in UCM pathogenesis" should be "in the pathogenesis of UCM", and etc.

Answer: Thanks for your constructive comments. “in UCM pathogenesis” has been amended to “in the pathogenesis of UCM” in the revised manuscript (Line 26 in the tracked copy). Moreover, we have restructured other sentences in the revised manuscript (Line 136-137, 163-166 in the tracked copy).

3) Define abbreviations when they appear for the first time and then use them when they appear in subsequent text. The abbreviation table can be included so that there is no need to define abbreviation in every figure legend and text, again.

Answer: Thank you for carefully and patiently reviewing our manuscript. We agree with you that abbreviations are defined when they appear for the first time and are then used when they appear in subsequent text. Considering the manuscript format of the journal, we have deleted the abbreviations in the figure legends and table note (Line 684, 686, 697-699, 710-713, 723-726, 739-742, 761-765, 782-787 in the tracked copy).

4) Define the abbreviation of bacteria, such as *K. pneumoniae* for *Klebsiella pneumoniae*, and then continue to use it throughout text. Authors sometimes used *K. pneumoniae* or *Klebsiella* or *pneumoniae*.

Answer: Thanks for raising this important issue. We have defined the abbreviation of the bacteria as "*K. pneumoniae*" and have then continued to use it throughout text (Line 32, 99, 101-103, 384-385, 760-761 in the tracked copy).

6) Be consistent in describing the *Bifidobacterium animalis*, such as *B. animalis* throughout text.

Answer: Thanks for raising this important issue. We have defined the abbreviation of *Bifidobacterium animalis* as "*B. animalis*" and have then used it throughout text (Line 184, 189, 385-386, 388-389, 392, 402 in the tracked copy).

7) Correct typos and grammatical mistakes, such as line 351, add comma after *animlalis* before respectively.

Answer: We appreciate you for raising the important issue. Following your advice, we have added comma after *animlalis* before respectively in the sentence in line 351 (Line 392 in the tracked copy). Moreover, we have corrected other typos and grammatical mistakes throughout text (Line 25, 93, 194, 222, 270, 284-286, 342-343, 418 in the tracked copy).

8) Please check line 703. I do not know the meaning of "whiskers" in this context?

Answer: Thank you for carefully and patiently reviewing our manuscript. In fact, Box-whisker Plot is also known as Boxplot. Whisker is the vertical line above and below the Box and is widely used in papers by authors (e.g., PMID: 35746334, PMID: 35450958, PMID: 34088052).

9) Define probiotic-gavage and bacterial-gavage in the method and use them throughout text. I assume that they are same experimental group.

Answer: Thanks for your careful review and insightful comments. In our study, administration of experimental compounds (e.g. probiotic suspension, *K. pneumoniae* suspension, stool microbiota suspension) to the mice was achieved by using the gavage technique. Gavage is the introduction of a compound into animal's stomach using a special feeding needle. The definition and related statements have been added to the method section of the revised manuscript (Line 359-361, 673-675 in the tracked copy).

Reviewer #2 (Comments for the Author):

This manuscript describes the pathogenic role for cardiac Th1 cells in Uremic Cardiomyopathy and the elevated abundance of opportunistic pathogen *Klebsiella pneumoniae* in CKD, and role of gut microbiota (*K. pneumoniae* in particular), and its contribution to the development of UCM via the induction of heart-infiltrating Th1 cells expansion. According to the authors' findings, they proposed that manipulation of the gut microbiome might serve as a promising target for the amelioration of UCM. Overall, this study is interesting and important for the UCM study and therapy. The manuscript is well organized and the data interpretation is convincing. However, some specific comments should be addressed.

Specific comments:

L28: Please insert “gene” before “sequencing”.

Answer: Thanks for your careful review and constructive comment. Following your advice, we have inserted “gene” before “sequencing” in the revised manuscript (Line 29 in the tracked copy).

L80: “...is strongly correlated to...”

Answer: We appreciate you for raising the important issue. We have corrected the sentence in the revised manuscript (Line 80 in the tracked copy).

L81-82: Please delete “Journal of American Society of Nephrology”.

Answer: Thanks for your constructive comment. We have deleted “Journal of American Society of Nephrology” in the revised manuscript (Line 81-82 in the tracked copy).

L87: Gut microbiome is highly dynamic. Changes of gut microbiome can cause gut dysbiosis but is NOT termed gut dysbiosis. The authors should revise this statement carefully.

Answer: Thanks for your careful review and insightful comment. We agree with you that changes of gut microbiome can cause gut dysbiosis but is NOT termed gut dysbiosis. Hence, we have deleted the inappropriate definition of gut dysbiosis in the revised manuscript (Line 88-89 in the tracked copy).

L91: “Emerging studies have...”

Answer: We appreciate you for raising the important issue. We have corrected the sentence in the revised manuscript (Line 93 in the tracked copy).

L92-93: The Enterobacteriaceae is a large family of bacteria, including many familiar pathogens, such as the genera Salmonella, Shigella and Escherichia. Klebsiella is also a genus of it. Which members of Enterobacteriaceae have been changed? The authors should make it clear.

Answer: Thanks for your careful review and insightful comments. We couldn't agree more with your standpoints. In fact, patients with CKD had gut dysbiosis characterized by enriched *K. pneumoniae* and other *Enterobacteriaceae* (e.g., *Escherichia*, *Shigella*, *Salmonella*). We have made the statement clear in the revised manuscript (Line 95-96 in the tracked copy).

L93: Please revise “Klebsiella are” to “Members of Klebsiella are”.

Answer: Thank you for raising the important issue. We have revised “*Klebsiella* are” to “Members of *Klebsiella* are” in the revised manuscript (Line 96 in the tracked copy).

L106-108: As shown in Figure 1A, the significantly decreased microbial diversity of CKD group compared to the control should be stated here.

Answer: Thank you for carefully and patiently reviewing our manuscript. In original manuscript, we stated “we found that fecal microbial composition and alpha-diversity of CKD patients differed from that of healthy controls”, which

included the information on difference in microbial diversity between the two groups.

L113-121: The purpose of this part is to show the experimental design and validity of FMT technique used?

Answer: Thank you for carefully and patiently reviewing our manuscript. Yes, the purpose of this part is to show the experimental design and validity of FMT technique used.

L123-125: Repeat of L113-114.

Answer: Thank you very much for pointing out the problem. We have deleted the part in the revised manuscript (Line 128-130 in the tracked copy).

L125-127: Figures 3C and 3D showed no significant differences between the four groups studied. Please check it.

Answer: Thank you for carefully and patiently reviewing our manuscript. We are very sorry that we did not state the part clearly in the results section of original manuscript. In fact, we found that CKD recipients developed more severe diastolic dysfunction compared to healthy controls (HC) recipients or control mice (Fig. 3, A-B). There were no significant differences in terms of EF and diastolic left ventricle anterior wall thickness (LVAW;d) between the four groups studied (Fig. 3, C-D). We have made the part clear in the revised manuscript (Line 132-134 in the tracked copy).

L127-130: The presentation for Figures 3E-3H is problematic. For correlations,

no differences between CKD and HC recipients, 3E vs 3G (both negative correlations), 3F vs 3H (both positive correlations). Please check it.

Answer: Thank you for carefully and patiently reviewing our manuscript. In our study, there were significant differences in regarding to E/A ration and MPI between CKD and HC recipients (Fig. 3, A-B).

Diastolic dysfunction is characterized by reduced E/A ration or (and) increased MPI. Figure 3E and 3G showed that the relative abundance of *K. pneumoniae* was negatively correlated with E/A ration in CKD recipients and HC recipients. Figure 3F and 3H showed that the relative abundance of *K. pneumoniae* was positively correlated with MPI in CKD recipients and HC recipients. Therefore, Figures 3E-3H showed that the relative abundance of *K. pneumoniae* was positively correlated with diastolic dysfunction in CKD recipients and HC recipients. Additionally, what need to be pointed out is that “recipient mice with CKD” has been amended to “CKD recipients and HC recipients” in the revised manuscript (Line 136-137 in the tracked copy).

L146-148: Except *K. pneumoniae*, any other microbial taxa showed similar correlation patterns?

Answer: Thanks for your careful review. In this study, no correlation was observed between abundance of other microbial taxa and levels of cardiac IFN γ ⁺ CD4⁺ T cells, which has been added to the result section of the revised manuscript (Line 158-159 in the tracked copy).

L152: “To test ...”

Answer: We appreciate you for raising the important issue. We have corrected the sentence in the revised manuscript (Line 163-166 in the tracked copy).

L152-154: Please revise this part to make better sense of statement.

Answer: Thanks for your careful review and insightful comment. To make better sense of the statement, we have amended it to “To test the effect of Th1-inducing bacteria on UCM, we gavaged *K. pneumoniae* to mouse model of CKD” in the revised manuscript (Line 163-166 in the tracked copy).

L158: Figure 5C, what's the pattern if PcoA used?

Answer: Thank you for carefully and patiently reviewing our manuscript. It showed similar pattern if PcoA used. We presented the result of NMDS in our manuscript.

L161: Figures 5G and 5H showed no significant differences between the two groups.

Answer: Thank you for carefully and patiently reviewing our manuscript. We are very sorry that we did not state the part clearly in the results section of original manuscript. In fact, we found that mice gavaged with *K. pneumoniae* showed significantly increased cardiac IFN γ ⁺ CD4⁺ T cells infiltration compared to the control mice (Fig. 5D), concomitantly with more severe diastolic dysfunction (Fig. 5, E-F). There were no significant differences in terms of EF and LVAW;d between the two groups (Fig. 5, G-H). We have made this part clear in the revised manuscript (Line 172-173 in the tracked copy).

L174: For Figure 6C, please see the comment of Figure 5C.

Answer: Thank you for carefully and patiently reviewing our manuscript. It showed similar pattern if PcoA used. We presented the result of NMDS in our manuscript.

L177: Figures 6G and 6H showed no significant differences between the two groups.

Answer: Thank you for carefully and patiently reviewing our manuscript. We are very sorry that we did not state the part clearly in the results section of original manuscript. In fact, we found that diastolic dysfunction was significantly improved in mice gavaged with *B. animalis* compared to the control mice (Fig. 6, E-F). There were no significant differences in terms of EF and LVAW;d between the two groups (Fig. 6, G-H). We have made this part clear in the revised manuscript (Line 191-192 in the tracked copy).

L178: "...two studied groups..."

Answer: We appreciate you for raising the important issue. We have corrected the sentence in the revised manuscript (Line 194 in the tracked copy).

L204: "...by a clinical study..."

Answer: Thank you very much for pointing out the problem. We have corrected the sentence in the revised manuscript (Line 220 in the tracked copy).

L255-257: "we believe that administration of antibiotic did not affect the observed differences in immune". This statement doesn't support your current

hypothesis and test. Antibiotics do affect immune because they can change gut microbiota largely. Please modify this statement carefully.

Answer: Thanks for your careful review and insightful comment. We agree with you that antibiotics do affect immune because they can change gut microbiota largely. We have deleted the inappropriate statement in the revised manuscript (Line 288-290 in the tracked copy).

L270: "...16S ribosomal RNA (rRNA) gene sequencing."

Answer: Thanks a lot for raising the important issue. We have added "gene" in the sentence in the revised manuscript (Line 304 in the tracked copy).

L302: "...based on a paired-end sequencing mode"

Answer: We appreciate you for pointing out the important issue. We have corrected the statement in the revised manuscript (Line 336 in the tracked copy).

L304: Duplicate sequences should be removed for shotgun sequencing but NOT for amplicon sequencing.

Answer: Thank you for your careful review. We agree with you. We have deleted "chimera removal" from the sentence in the revised manuscript (Line 339 in the tracked copy).

L303-312: Please insert citations for these tools.

Answer: Thanks for your valuable recommendation. We have inserted citations for these tools in the revised manuscript (Line 339-340, 342, 656-664 in the tracked copy).

L344-351: Please use abbreviations for “Bifidobacterium animalis”. And other places in the manuscript.

Answer: Thanks for your careful review. Following your advice, we have defined the abbreviation of the *Bifidobacterium animalis* as “*B. animalis*” and have used it throughout text (Line 184, 189, 385-386, 388-389, 392, 402 in the tracked copy).

L377: ...stain incubated...?

Answer: Thanks for your careful review. We are very sorry that we made a clerical error. We have deleted “stain” from the sentence in the revised manuscript (Line 418 in the tracked copy).

L411: “National Natural Science Foundation of China”

Answer: We appreciate you for raising the important issue. We have corrected the statement in the revised manuscript (Line 452 in the tracked copy).

Staff Comments:

Preparing Revision Guidelines

- Point-by-point responses to the issues raised by the reviewers in a file named "Response to Reviewers," NOT IN YOUR COVER LETTER.
- Upload a compare copy of the manuscript (without figures) as a "Marked-Up Manuscript" file.
- Each figure must be uploaded as a separate file, and any multipanel figures must be assembled into one file.
- Manuscript: A .DOC version of the revised manuscript
- Figures: Editable, high-resolution, individual figure files are required at revision, TIFF or EPS files are preferred.

Answer: Thanks for your careful review and critical comments. We have accomplished all the requirements in the list above.

November 7, 2022

Prof. Weijie Yuan
Shanghai General Hospital
85 Wujin Road
Shanghai
China

Re: Spectrum03101-22R1 (**Dysbiosis of Gut Microbiota Contributes to Uremic Cardiomyopathy via Induction of IFN γ -producing CD4 $^+$ T Cells Expansion**)

Dear Prof. Weijie Yuan:

Link Not Available

Sincerely,

Pei-Yuan Qian

Journals Department
Reviewer comments:

Reviewer #1 (Comments for the Author):

The revised version of the manuscript addressed my previous comments and concerns.

Reviewer #2 (Comments for the Author):

L323-325: Duplicate sequences should be removed for shotgun sequencing but NOT for amplicon sequencing. The authors did not get my point. For 16S amplicon sequencing, duplicate sequences should be kept to determine relative

abundance. Low-quality and chimeric sequences must be removed before OTU picking. The authors can revise this statement as "...by UPARSE filtering of low-quality and chimeric sequences".

Staff Comments:

Preparing Revision Guidelines

Please return the manuscript within 60 days; if you cannot complete the modification within this time period, please contact me. If you do not wish to modify the manuscript and prefer to submit it to another journal, please notify me of your decision immediately so that the manuscript may be formally withdrawn from consideration by Microbiology Spectrum.

Response to Reviewer Comments

Reviewer #1 (Comments for the Author):

The revised version of the manuscript addressed my previous comments and concerns.

Answer: We are truly grateful to the positive comments from you.

Reviewer #2 (Comments for the Author):

L323-325: Duplicate sequences should be removed for shotgun sequencing but NOT for amplicon sequencing. The authors did not get my point. For 16S amplicon sequencing, duplicate sequences should be kept to determine relative abundance. Low-quality and chimeric sequences must be removed before OTU picking. The authors can revise this statement as "...by UPARSE filtering of low-quality and chimeric sequences".

Answer: Thank you for carefully and patiently reviewing our manuscript. Following your advice, we have revised this statement as "...by UPARSE filtering of low-quality and chimeric sequences" in the method section of the revised manuscript (Line 326-327 in the tracked copy).

Staff Comments:

- Point-by-point responses to the issues raised by the reviewers in a file named "Response to Reviewers," NOT IN YOUR COVER LETTER.

- Upload a compare copy of the manuscript (without figures) as a "Marked-Up Manuscript" file.
- Each figure must be uploaded as a separate file, and any multipanel figures must be assembled into one file.
- Manuscript: A .DOC version of the revised manuscript
- Figures: Editable, high-resolution, individual figure files are required at revision, TIFF or EPS files are preferred.

Answer: Thanks for your careful review and critical comments. We have accomplished all the requirements in the list above.

November 18, 2022

Prof. Weijie Yuan
Shanghai General Hospital
85 Wujin Road
Shanghai
China

Re: Spectrum03101-22R2 (**Dysbiosis of Gut Microbiota Contributes to Uremic Cardiomyopathy via Induction of IFN γ -producing CD4 $^+$ T Cells Expansion**)

Dear Prof. Weijie Yuan:

The authors have addressed concerns raised by the reviewers and the paper is accepted for publication now.

Your manuscript has been accepted, and I am forwarding it to the ASM Journals Department for publication. You will be notified when your proofs are ready to be viewed.

Sincerely,

Pei-Yuan Qian
Editor, Microbiology Spectrum
